# Testing for Families of Distributions via the Fourier Transform[*]

**Clément L. Canonne**
Stanford University
ccanonne@stanford.edu

**Ilias Diakonikolas**
University of Southern California
diakonik@usc.edu

**Alistair Stewart**
University of Southern California
stewart.al@gmail.com

## Abstract

We study the general problem of testing whether an unknown discrete distribution belongs to a specified family of distributions. More specifically, given a distribution family $\mathcal{P}$ and sample access to an unknown discrete distribution $\mathbf{P}$, we want to distinguish (with high probability) between the case that $\mathbf{P} \in \mathcal{P}$ and the case that $\mathbf{P}$ is $\epsilon$-far, in total variation distance, from every distribution in $\mathcal{P}$. This is the prototypical hypothesis testing problem that has received significant attention in statistics and, more recently, in computer science. The main contribution of this work is a simple and general testing technique that is applicable to all distribution families whose *Fourier spectrum* satisfies a certain approximate *sparsity* property. We apply our Fourier-based framework to obtain near sample-optimal and computationally efficient testers for the following fundamental distribution families: Sums of Independent Integer Random Variables (SIIRVs), Poisson Multinomial Distributions (PMDs), and Discrete Log-Concave Distributions. For the first two, ours are the first non-trivial testers in the literature, vastly generalizing previous work on testing Poisson Binomial Distributions. For the third, our tester improves on prior work in both sample and time complexity.

## 1 Introduction

### 1.1 Background and Motivation

The prototypical inference question in the area of *distribution property testing* [6] is the following: Given a set of samples from a collection of probability distributions, can we determine whether these distributions satisfy a certain property? During the past two decades, this broad question – whose roots lie in statistical hypothesis testing [43, 40] – has received considerable attention by the computer science community, see [49, 10] for two recent surveys. After two decades of study, for many properties of interest there exist sample-optimal testers (matched by information-theoretic lower bounds) [44, 14, 54, 30, 2, 29, 12].

In this work, we focus on the problem of testing whether an unknown distribution belongs to a given family of discrete *structured* distributions. Let $\mathcal{P}$ be a family of discrete distributions over a total order (e.g., $[n]$) or a partial order (e.g., $[n]^k$). The problem of *membership testing for $\mathcal{P}$* is the following: Given sample access to an unknown distribution $\mathbf{P}$ (effectively supported on the same domain as $\mathcal{P}$), we want to distinguish between the case that $\mathbf{P} \in \mathcal{P}$ versus $d_{\mathrm{TV}}(\mathbf{P}, \mathcal{P}) > \epsilon$. (Here, $d_{\mathrm{TV}}$ denotes the

total variation distance between distributions.) The sample complexity of this problem depends on the underlying family $\mathcal{P}$. For example, if $\mathcal{P}$ contains a single distribution over a domain of size $n$, the sample complexity of the testing problem is $O(n^{1/2}/\epsilon^2)$ [14, 54, 30, 2].

In this work, we give a general technique to test membership in various distribution families over discrete domains, i.e., to solve the following task:

---

$\mathfrak{T}(\mathcal{P}, \epsilon)$: given a family of discrete distributions $\mathcal{P}$ over some partially or totally ordered set, parameter $\epsilon \in (0, 1]$, and sample access to an unknown distribution $\mathbf{P}$ over the same domain, how many samples are required to distinguish, with probability $3/5$, between the case that $\mathbf{P} \in \mathcal{P}$ versus $d_{\mathrm{TV}}(\mathbf{P}, \mathcal{P}) > \epsilon$?

---

Before we state our results in full generality, we present concrete applications to a number of well-studied distribution families.

## 1.2 Our Results

Our first result is a nearly sample-optimal testing algorithm for sums of independent integer random variables (SIIRVs). Formally, an $(n, k)$-SIIRV is a sum of $n$ independent integer random variables each supported in $\{0, 1, \ldots, k-1\}$. We will denote the set of $(n, k)$-SIIRVs by $\mathcal{SIIRV}_{n,k}$. SIIRVs comprise a rich class of distributions that arise in many settings. The special case of $k = 2$ was first considered by Poisson [45] as a non-trivial extension of the Binomial distribution, and is known as Poisson binomial distribution (PBD). In application domains, SIIRVs have many uses in research areas such as survey sampling, case-control studies, and survival analysis, see e.g., [16] for a survey of the many practical uses of these distributions. In addition to their practical applications, SIIRVs are of fundamental probabilistic interest and have been extensively studied in the theory of probability and statistics [19, 38, 34, 46, 39, 5, 18, 15]. We prove:

**Theorem 1** (Testing SIIRVs). *Given parameters $k, n \in \mathbb{N}$ and sample access to a distribution over $\mathbb{N}$, there exists an algorithm (Algorithm 1) for $\mathfrak{T}(\mathcal{SIIRV}_{n,k}, \epsilon)$ which takes*

$$O\left(\frac{kn^{1/4}}{\epsilon^2} \log^{1/4} \frac{1}{\epsilon} + \frac{k^2}{\epsilon^2} \log^2 \frac{k}{\epsilon}\right)$$

*samples, and runs in time $n(k/\epsilon)^{O(k \log(k/\epsilon))}$.*

Prior to our work, no non-trivial[2] tester was known for $(n, k)$-SIIRVs for any $k > 2$. [11] showed a sample lower bound of $\Omega\big(k^{1/2}n^{1/4}/\epsilon^2\big)$, but their techniques did not yield any non-trivial sample upper bound.

For the special case of PBDs ($k = 2$), Acharya and Daskalakis [1] gave a tester with sample complexity $O\left(\frac{n^{1/4}}{\epsilon^2}\sqrt{\log 1/\epsilon} + \frac{\log^{5/2} 1/\epsilon}{\epsilon^6}\right)$, running time $O\left(\frac{n^{1/4}}{\epsilon^2}\sqrt{\log 1/\epsilon} + (1/\epsilon)^{O(\log^2 1/\epsilon)}\right)$, and also showed a sample lower bound of $\Omega(n^{1/4}/\epsilon^2)$. The special case of our Theorem 1 for $k = 2$ yields an improvement over [1] in both sample size and runtime:

**Theorem 2** (Testing PBDs). *Given parameter $n \in \mathbb{N}$ and sample access to a distribution over $\mathbb{N}$, there exists an algorithm (Algorithm 1) for $\mathfrak{T}(\mathcal{PBD}_n, \epsilon)$ which takes*

$$O\left(\frac{n^{1/4}}{\epsilon^2} \log^{1/4} \frac{1}{\epsilon} + \frac{\log^2 1/\epsilon}{\epsilon^2}\right)$$

*samples, and runs in time $n^{1/4} \cdot \tilde{O}\big(1/\epsilon^2\big) + (1/\epsilon)^{O(\log \log(1/\epsilon))}$.*

Note that the sample complexity of our algorithm is $n^{1/4} \cdot \tilde{O}(1/\epsilon^2)$, matching the information-theoretic lower bound up to a logarithmic factor in $1/\epsilon$. In particular, our algorithm does not incur the extraneous $\Omega(1/\epsilon^6)$ term of [1]. Moreover, our runtime has a $(1/\epsilon)^{O(\log \log(1/\epsilon))}$ dependence, as opposed to $(1/\epsilon)^{O(\log^2 1/\epsilon)}$. The improved running time relies on a more efficient computational

"projection step" in our general framework, which leverages the geometric structure of Poisson Binomial distributions.

We remark that the guarantees provided by the above two theorems are actually stronger than the usual property testing one. Namely, whenever the algorithm returns **accept**, then it also provides a (proper) hypothesis $\mathbf{H}$ such that $d_{\mathrm{TV}}(\mathbf{P}, \mathbf{H}) \leq \epsilon$ with probability at least $3/5$.

A broad generalization of PBDs to the high-dimensional setting is the family of Poisson Multinomial Distributions (PMDs). Formally, an $(n,k)$-PMD is any random variable of the form $X = \sum_{i=1}^{n} X_i$, where the $X_i$'s are independent random vectors supported on the set $\{e_1, e_2, \ldots, e_k\}$ of standard basis vectors in $\mathbb{R}^k$. We will denote by $\mathcal{PMD}_{n,k}$ the set of $(n,k)$-PMDs. PMDs comprise a broad class of discrete distributions of fundamental importance in computer science, probability, and statistics. A large body of work in the probability and statistics literature has been devoted to the study of the behavior of PMDs under various structural conditions [4, 41, 5, 7, 47, 48]. PMDs generalize the familiar multinomial distribution, and describe many distributions commonly encountered in computer science (see, e.g., [25, 26, 56, 53]). Recent years have witnessed a flurry of research activity on PMDs and related distributions, from several perspectives of theoretical computer science, including learning [22, 21, 31, 23, 32], property testing [56, 52, 53], computational game theory [25, 26, 9, 24, 27, 35, 17], and derandomization [37, 8, 28, 36]. We prove the following:

**Theorem 3** (Testing PMDs). *Given parameters $k, n \in \mathbb{N}$ and sample access to a distribution over $\mathbb{N}^k$, there exists an algorithm for $\mathfrak{T}(\mathcal{PMD}_{n,k}, \epsilon)$ which takes*

$$O\left(\frac{n^{(k-1)/4}k^{2k}}{\epsilon^2} \log(k/\epsilon)^k\right)$$

*samples, and runs in time $n^{O(k^3)} \cdot (1/\epsilon)^{O(k^3 \frac{\log(k/\epsilon)}{\log\log(k/\epsilon)})^{k-1}}$ or alternatively in time $n^{O(k)} \cdot 2^{O(k^{5k}\log(1/\epsilon)^{k+2})}$.*

For the sake of intuition, we note that Theorem 3 is particularly interesting in the regime that $n$ is large and $k$ is small. Indeed, the sample complexity of testing PMDs is inherently *exponential* in $k$: We prove a sample lower bound of $\Omega_k(n^{(k-1)/4}/\epsilon^2)$ (Theorem 8),[3] nearly-matching our upper bound for constant $k$.

Finally, we demonstrate the versatility of our techniques by obtaining a testing algorithm for discrete log-concavity. Log-concave distributions constitute a broad and flexible non-parametric family that is extensively used in modeling and inference [57]. In the discrete setting, log-concave distributions encompass a range of fundamental types of discrete distributions, including binomial, negative binomial, geometric, hypergeometric, Poisson, Poisson Binomial, hyper-Poisson, Pólya-Eggenberger, and Skellam distributions. Log-concave distributions have been studied in a wide range of different contexts including economics [3], statistics and probability theory (see [50] for a recent survey), theoretical computer science [42], and algebra, combinatorics and geometry [51]. We will denote by $\mathcal{LCV}_n$ the class of log-concave distributions over $[n]$. We prove:

**Theorem 4** (Testing Log-Concavity). *Given a parameter $n \in \mathbb{N}$ and sample access to a distribution over $\mathbb{N}$, there exists an algorithm for $\mathfrak{T}(\mathcal{LCV}_n, \epsilon)$ which takes*

$$O\left(\frac{\sqrt{n}}{\epsilon^2}\right) + \tilde{O}\left(\frac{1}{\epsilon^{5/2}}\right)$$

*samples, and runs in time $O(\sqrt{n} \cdot \mathrm{poly}(1/\epsilon))$.*

Our discrete log-concavity tester improves on previous work in terms of both sample and time complexity. Specifically, [2] gave a log-concavity tester with sample complexity $O(\sqrt{n}/\epsilon^2 + 1/\epsilon^5)$, while [11] obtained a tester with sample complexity $\tilde{O}(\sqrt{n}/\epsilon^{7/2})$. Our sample complexity dominates both these bounds, and is significantly better when $\epsilon$ is small. The algorithms in [2, 11] run in $\mathrm{poly}(n/\epsilon)$ time, as they involve solving a linear program of $\mathrm{poly}(n/\epsilon)$ size. In contrast, the running time of our algorithm is *sublinear* in $n$.

## 1.3 Our Techniques and Comparison to Previous Work

All the testing algorithms in this paper follow from a simple and general technique that may be of broader interest. The common property of the underlying distribution families $\mathcal{P}$ that allows for our unified testing approach is the following: Let $\mathbf{P}$ be the probability mass function of any distribution in $\mathcal{P}$. Then, *the Fourier transform of $\mathbf{P}$ is approximately sparse*, in a well-defined sense.

For concreteness, we elaborate on our technique for the case of SIIRVs. The starting point of our approach is the observation from [31] that $(n, k)$-SIIRVs – in addition to having a relatively small effective support – also have an approximately sparse Fourier representation. Roughly speaking, most of their Fourier mass is concentrated on a small subset of Fourier coefficients, which can be computed efficiently.

This suggests the following natural approach to testing $(n, k)$-SIIRVs: first, identify the effective support $I$ of the distribution $\mathbf{P}$ and check that it is appropriately small. If it is not, then reject. Then, compute the corresponding small subset $S$ of the Fourier domain, and check that almost no Fourier mass of $\mathbf{P}$ lies outside $S$. Otherwise, one can safely reject, as this is a certificate that $\mathbf{P}$ is not an $(n, k)$-SIIRV. Combining the two steps, one can show that learning the Fourier transform of $\mathbf{P}$ (in $L_2$-norm) on this small subset $S$ only, is sufficient to learn $\mathbf{P}$ itself in total variation distance. The former goal can be performed with relatively few samples, as $S$ is sufficiently small.

At this point, we have obtained a distribution $\mathbf{H}$ – succinctly represented by its Fourier transform on $S$ – such that $\mathbf{P}$ and $\mathbf{H}$ are close in total variation distance. It only remains to perform a computational "projection step" to verify that $\mathbf{H}$ itself is close to some $(n, k)$-SIIRV. This will clearly be the case if indeed $\mathbf{P} \in \mathcal{SIIRV}_{n,k}$.

Although the aforementioned approach forms the core of our SIIRV testing algorithm, the actual tester has to address separately the case where $\mathbf{P}$ has small variance, which can be handled by a testing-via-learning approach. Our main contribution is thus to describe how to efficiently perform the second step, i.e., the Fourier sparsity testing. This is done in Theorem 6, which describes a simple algorithm to perform this step. The algorithm proceeds by essentially considering the Fourier coefficients of the empirical distribution (obtained by taking a small number of samples). Interestingly, the main idea underlying Theorem 6 is to avoid analyzing directly the behavior of these Fourier coefficients – which would naively require too high a time complexity. Instead, we rely on Plancherel's identity and reduce the problem to the analysis of a different task: that of the sample complexity of $L_2$ *identity testing* (Proposition 1). By a tight analysis of this $L_2$ tester, we get as a byproduct that several Fourier quantities of interest (of our empirical distribution) simultaneously enjoy good concentration – while arguing concentration of each of these terms separately would yield a suboptimal time complexity.

A nearly identical method works for PMDs as well. Moreover, our approach can be abstracted to yield a general testing framework, as we explain in Section 4. It is interesting to remark that the Fourier transform has been used to learn PMDs and SIIRVs [31, 23, 32, 20], and therefore it may not be entirely surprising that it has applications to testing as well. Notably, our Fourier testing technique gives an improved and nearly-optimal algorithms for log-concavity, for which no Fourier learning algorithm was known. More generally, testing membership to a class using the Fourier transform is significantly more challenging than learning. A fundamental difference is that in the testing setting we need to handle distributions that do *not* belong to the class (e.g., SIIRVs, PMDs), but are far from the class in an arbitrary way. In contrast, learning algorithms work under the promise that the distribution is in the underlying class, and thus can leverage the specific structure.

**Testing via the Fourier Transform: the Advantage**   One may wonder how the detour via the Fourier transform enables us to obtain better sample complexity than an approach purely based on $L_2$ testing. Indeed, all distributions in the classes we consider, crucially, have small $L_2$ norm. For testing identity to such a distribution $\mathbf{P}$, the standard $L_2$ identity tester (see, e.g., [14] or Proposition 1), which works by checking how large the $L_2$-distance between the empirical and the hypothesis distribution is, will be optimal. We can thus test membership of a class of such distributions by (i) learning $\mathbf{P}$ assuming it belongs to the class, and then (ii) test whether what we learned is indeed close to $\mathbf{P}$ using the $L_2$ identity tester. The catch is that, in order to get guarantees in $L_1$-distance using this approach, would require us to learn to very small $L_2$ distance (because of the Cauchy–Schwarz inequality). In particular, if the unknown distribution $\mathbf{P}$ has support size $N$, we would have to learn to $L_2$ distance $\epsilon/\sqrt{N}$ in (i), and then in (ii) test that we are within $L_2$-distance $\epsilon/\sqrt{N}$ of the learned hypothesis.

However, if a distribution $\mathbf{P}$ has a sparse discrete Fourier transform (whose effective support is known), then it suffices to estimate only these few Fourier coefficients [31, 33]. This step enables us to learn $\mathbf{P}$ in (i) not just to within $L_1$-distance $\epsilon$, but indeed (crucially) within $L_2$-distance $\frac{\epsilon}{\sqrt{N}}$ with good sample complexity. Additionally, the identity testing algorithm can be put into a simpler form for a hypothesis with sparse Fourier transform, as previously mentioned. Now, the tester has higher sample complexity, roughly $\sqrt{N}/\epsilon^2$; but if it accepts, then we have learned the distribution $\mathbf{P}$ to within $\epsilon$ total variation distance, with much fewer samples than the $\Omega(N/\epsilon^2)$ required for arbitrary distributions over support size $N$. Lastly, we note that we can replace the support size $N$ in the above description by the size of the *effective support*, i.e., the smallest set that contains $1 - O(\epsilon)$ fraction of the mass. Doing so for the case of $(n, k)$-SIIRVs leads to a sample complexity proportional to $n^{1/4}$, instead of $n^{1/2}$.

## 1.4 Organization

The rest of the paper is organized as follows: In Section 2, we set up notation and provide definitions as well as standard results relevant to our purposes. Section 3 contains the details of one of the main subroutines our testers rely on, namely for *Fourier sparsity testing*. In Section 4, we describe our general approach to obtain a tester applicable to any class of distributions which enjoys good Fourier sparsity. In Section 5, we state and sketch the proof of our sample complexity lower bound for testing PMDs. Due to space constraints, most proofs have been deferred to the full version [13].

## 2 Notation and Definitions

We begin with some standard notations and definitions, as well as basics of Fourier analysis and results from Probability that we shall use throughout the paper. For $m \in \mathbb{N}$, we write $[m]$ for the set $\{0, 1, \dots, m-1\}$, and $\log$ (resp. $\ln$) for the binary logarithm (resp. the natural logarithm).

**Distributions and Metrics** A probability distribution over (discrete) domain $\Omega$ is a function $\mathbf{P} \colon \Omega \to [0, 1]$ such that $\|\mathbf{P}\|_1 \overset{\text{def}}{=} \sum_{\omega \in \Omega} \mathbf{P}(\omega) = 1$; we denote by $\Delta(\Omega)$ the set of all probability distributions over domain $\Omega$. Recall that for two probability distributions $\mathbf{P}, \mathbf{Q} \in \Delta(\Omega)$, their *total variation distance* (or statistical distance) is defined as $d_{\mathrm{TV}}(\mathbf{P}, \mathbf{Q}) \overset{\text{def}}{=} \sup_{S \subseteq \Omega}(\mathbf{P}(S) - \mathbf{Q}(S)) = \frac{1}{2} \sum_{\omega \in \Omega} |\mathbf{P}(\omega) - \mathbf{Q}(\omega)|$, i.e. $d_{\mathrm{TV}}(\mathbf{P}, \mathbf{Q}) = \frac{1}{2} \|\mathbf{P} - \mathbf{Q}\|_1$. Given a subset $\mathcal{P} \subseteq \Delta(\Omega)$ of distributions, the *distance from $\mathbf{P}$ to $\mathcal{P}$* is then defined as $d_{\mathrm{TV}}(\mathbf{P}, \mathcal{P}) \overset{\text{def}}{=} \inf_{\mathbf{Q} \in \mathcal{P}} d_{\mathrm{TV}}(\mathbf{P}, \mathbf{Q})$. If $d_{\mathrm{TV}}(\mathbf{P}, \mathcal{P}) > \epsilon$, we say that $\mathbf{P}$ is *$\epsilon$-far* from $\mathcal{P}$; otherwise, it is *$\epsilon$-close*.

**Discrete Fourier Transform** Our algorithms will rely heavily on the (discrete) Fourier transform, whose definition we recall next.

**Definition 1** (Discrete Fourier Transform). For $x \in \mathbb{R}$, we let $e(x) \overset{\text{def}}{=} \exp(-2i\pi x)$. The *Discrete Fourier Transform (DFT) modulo $M$* of a function $F \colon [n] \to \mathbb{C}$ is the function $\widehat{F} \colon [M] \to \mathbb{C}$ defined as

$$\widehat{F}(\xi) = \sum_{j=0}^{n-1} e\left(\frac{\xi j}{M}\right) F(j)$$

for $\xi \in [M]$. The DFT modulo $M$ of a distribution $\mathbf{P}$, $\widehat{\mathbf{P}}$, is then the DFT modulo $M$ of its probability mass function (note that one can then equivalently see $\widehat{\mathbf{P}}(\xi)$ as the expectation $\widehat{\mathbf{P}}(\xi) = \mathbb{E}_{X \sim F}[e\left(\frac{\xi X}{M}\right)]$, for $\xi \in [M]$).

The *inverse DFT modulo $M$* onto the range $[m, m + M - 1]$ of $\widehat{F} \colon [M] \to \mathbb{C}$, is the function $F \colon [m, m + M - 1] \cap \mathbb{Z} \to \mathbb{C}$ defined by

$$F(j) = \frac{1}{M} \sum_{\xi=0}^{M-1} e\left(-\frac{\xi j}{M}\right) \widehat{F}(\xi),$$

for $j \in [m, m + M - 1] \cap \mathbb{Z}$.

Note that the DFT (modulo $M$) is a linear operator; moreover, we recall the standard fact relating the norms of a function and of its Fourier transform, that we will use extensively:

**Theorem 5** (Plancherel's Theorem). *For $M \geq n$ and $F, G \colon [n] \to \mathbb{C}$, we have (i) $\sum_{j=0}^{n-1} F(j)\overline{G(j)} = \frac{1}{M} \sum_{\xi=0}^{M-1} \widehat{F}(\xi)\overline{\widehat{G}(\xi)}$; and (ii) $\|F\|_2 = \frac{1}{\sqrt{M}}\|\widehat{F}\|_2$, where $\widehat{F}, \widehat{G}$ are the DFT modulo $M$ of $F, G$, respectively.*

(The latter equality is sometimes referred to as Parseval's theorem.) We also note that, for our PMD testing, we shall need the appropriate generalization of the Fourier transform to the multivariate setting. We leave this generalization to the full version.

## 3 Testing Effective Fourier Support

In this section, we prove the following theorem, which will be invoked as a crucial ingredient of our testing algorithms. Broadly speaking, the theorem ensures one can efficiently test whether an unknown distribution $\mathbf{Q}$ has its Fourier transform concentrated on some (small) effective support $S$ (and if this is the case, learn the vector $\widehat{\mathbf{Q}}\mathbf{1}_S$, the restriction of this Fourier transform to $S$, in $L_2$ distance).

**Theorem 6.** *Given parameters $M \geq 1$, $\epsilon, b \in (0, 1]$, as well as a subset $S \subseteq [M]$ and sample access to a distribution $\mathbf{Q}$ over $[M]$, Algorithm 1 outputs either reject or a collection of Fourier coefficients $\widehat{\mathbf{H}'} = (\widehat{\mathbf{H}'}(\xi))_{\xi \in S}$ such that with probability at least $7/10$, all the following statements hold simultaneously.*

   1. *if $\|\mathbf{Q}\|_2^2 > 2b$, then it outputs reject;*

   2. *if $\|\mathbf{Q}\|_2^2 \leq 2b$ and every function $\mathbf{Q}^* \colon [M] \to \mathbb{R}$ with $\widehat{\mathbf{Q}^*}$ supported entirely on $S$ is such that $\|\mathbf{Q} - \mathbf{Q}^*\|_2 > \epsilon$, then it outputs reject;*

   3. *if $\|\mathbf{Q}\|_2^2 \leq b$ and there exists a function $\mathbf{Q}^* \colon [M] \to \mathbb{R}$ with $\widehat{\mathbf{Q}^*}$ supported entirely on $S$ such that $\|\mathbf{Q} - \mathbf{Q}^*\|_2 \leq \frac{\epsilon}{2}$, then it does not output reject;*

   4. *if it does not output reject, then $\|\widehat{\mathbf{Q}}\mathbf{1}_S - \widehat{\mathbf{H}'}\|_2 \leq \frac{\epsilon\sqrt{M}}{10}$ and the inverse Fourier transform (modulo $M$) $\mathbf{H}'$ of the Fourier coefficients $\widehat{\mathbf{H}'}$ it outputs satisfies $\|\mathbf{Q} - \mathbf{H}'\|_2 \leq \frac{6\epsilon}{5}$.*

*Moreover, the algorithm takes $m = O\left(\frac{\sqrt{b}}{\epsilon^2} + \frac{|S|}{M\epsilon^2} + \sqrt{M}\right)$ samples from $\mathbf{Q}$, and runs in time $O(m|S|)$.*

Note that the rejection condition in Item 2 is equivalent to $\|\widehat{\mathbf{Q}}\mathbf{1}_{\bar{S}}\|_2 > \epsilon\sqrt{M}$, that is to having Fourier mass more than $\epsilon^2$ outside of $S$; this is because for any $\mathbf{Q}^*$ supported on $S$,

$$M\|\mathbf{Q} - \mathbf{Q}^*\|_2^2 = \|\widehat{\mathbf{Q}} - \widehat{\mathbf{Q}^*}\|_2^2 = \|\widehat{\mathbf{Q}}\mathbf{1}_S - \widehat{\mathbf{Q}^*}\mathbf{1}_S\|_2^2 + \|\widehat{\mathbf{Q}}\mathbf{1}_{\bar{S}} - \widehat{\mathbf{Q}^*}\mathbf{1}_{\bar{S}}\|_2^2 \geq \|\widehat{\mathbf{Q}}\mathbf{1}_{\bar{S}} - \widehat{\mathbf{Q}^*}\mathbf{1}_{\bar{S}}\|_2^2 = \|\widehat{\mathbf{Q}}\mathbf{1}_{\bar{S}}\|_2^2$$

and the inequality is tight for $\mathbf{Q}^*$ being the inverse Fourier transform (modulo $M$) of $\widehat{\mathbf{Q}}\mathbf{1}_S$.

**High-level idea.** Let $\mathbf{Q}$ be an unknown distribution supported on $M$ consecutive integers (we will later apply this to $\mathbf{Q} \overset{\text{def}}{=} \mathbf{P} \bmod M$), and $S \subseteq [M]$ be a set of Fourier coefficients (symmetric with regard to $M$: $\xi \in S$ implies $-\xi \bmod M \in S$) such that $0 \in S$. We can further assume that we know $b \geq 0$ such that $\|\mathbf{Q}\|_2^2 \leq b$.

Given $\mathbf{Q}$, we can consider its "truncated Fourier expansion" (with respect to $S$) $\widehat{\mathbf{H}} = \widehat{\mathbf{Q}}\mathbf{1}_S$ defined as

$$\widehat{\mathbf{H}}(\xi) \overset{\text{def}}{=} \begin{cases} \widehat{\mathbf{Q}}(\xi) & \text{if } \xi \in S \\ 0 & \text{otherwise} \end{cases}$$

for $\xi \in [M]$; and let $\mathbf{H}$ be the inverse Fourier transform (modulo $M$) of $\widehat{\mathbf{H}}$. Note that $\mathbf{H}$ is no longer in general a probability distribution.

To obtain the guarantees of Theorem 6, a natural idea is to take some number $m$ of samples from $\mathbf{Q}$, and consider the empirical distribution $\mathbf{Q}'$ they induce over $[M]$. By computing the Fourier coefficients (restricted to $S$) of this $\mathbf{Q}'$, as well as the Fourier mass "missed" when doing so (i.e., the Fourier mass $\|\widehat{\mathbf{Q}'}\mathbf{1}_{\bar{S}}\|_2^2$ that $\mathbf{Q}'$ puts outside of $S$) to sufficient accuracy, one may hope to prove Theorem 6 with a reasonable bound on $m$.

The issue is that analyzing *separately* the behavior of $\|\widehat{\mathbf{Q}'}\mathbf{1}_{\bar{S}}\|_2^2$ and $\|\widehat{\mathbf{Q}}\mathbf{1}_S - \widehat{\mathbf{Q}'}\mathbf{1}_S\|_2^2$ to show that they are both estimated sufficiently accurately, and both small enough, is not immediate. Instead, we will get a bound on both at the same time, by arguing concentration in a different manner – namely, by analyzing a different tester for tolerant identity testing in $L_2$ norm.

In more detail, letting $\mathbf{H}$ be as above, we have by Plancherel that

$$\sum_{i\in[M]}(\mathbf{Q}'(i)-\mathbf{H}(i))^2 = \|\mathbf{Q}'-\mathbf{H}\|_2^2 = \frac{1}{M}\|\widehat{\mathbf{Q}'}-\widehat{\mathbf{H}}\|_2^2 = \frac{1}{M}\sum_{\xi=0}^{M-1}|\widehat{\mathbf{Q}'}(\xi)-\widehat{\mathbf{H}}(\xi)|^2$$

and, expanding the definition of $\widehat{\mathbf{H}}$ and using Plancherel again, this can be rewritten as

$$M\sum_{i\in[M]}(\mathbf{Q}'(i)-\mathbf{H}(i))^2 = \|\widehat{\mathbf{Q}}\mathbf{1}_S - \widehat{\mathbf{Q}'}\mathbf{1}_S\|_2^2 + M\|\mathbf{Q}'\|_2^2 - \|\widehat{\mathbf{Q}'}\mathbf{1}_S\|_2^2.$$

(The full derivation will be given in the proof.) The right-hand side has two non-negative compound terms: the first, $\|\widehat{\mathbf{Q}}\mathbf{1}_S - \widehat{\mathbf{Q}'}\mathbf{1}_S\|_2^2$, corresponds to the $L_2$ error obtained when learning the Fourier coefficients of $\mathbf{Q}$ on $S$. The second, $M\|\mathbf{Q}'\|_2^2 - \|\widehat{\mathbf{Q}'}\mathbf{1}_S\|_2^2 = \|\widehat{\mathbf{Q}'}\mathbf{1}_{\bar{S}}\|_2^2$, is the Fourier mass that our empirical $\mathbf{Q}'$ puts "outside of $S$."

So if the LHS is small (say, order $\epsilon^2$), then in particular both terms of the RHS will be small as well, effectively giving us bounds on our two quantities in one shot. But this very same LHS is very reminiscent of a known statistic [14] for testing identity of distributions in $L_2$. So, one can analyze the number of samples required by analyzing such an $L_2$ tester instead. This is what we will do in Proposition 1.

---

**Algorithm 1** Testing the Fourier Transform Effective Support

---

**Require:** parameters $M\geq 1$, $b,\epsilon\in(0,1]$; set $S\subseteq[M]$; sample access to distribution $\mathbf{Q}$ over $[M]$
  1: Set $m\leftarrow\left\lceil C(\frac{\sqrt{b}}{\epsilon^2}+\frac{|S|}{M\epsilon^2}+\sqrt{M})\right\rceil$          $\triangleright$ $C>0$ is an absolute constant
  2: Draw $m'\leftarrow\mathrm{Poi}(m)$; if $m'>2m$, **return** reject
  3: Draw $m'$ samples from $\mathbf{Q}$, and let $\mathbf{Q}'$ be the corresponding empirical distribution over $[M]$
  4: Compute $\|\mathbf{Q}'\|_2^2$, $\widehat{\mathbf{Q}'}(\xi)$ for every $\xi\in S$, and $\|\widehat{\mathbf{Q}'}\mathbf{1}_S\|_2^2$          $\triangleright$ Takes time $O(m|S|)$
  5: **if** $m'^2\|\mathbf{Q}'\|_2^2 - m' > \frac{3}{2}bm^2$ **then return** reject
  6: **else if** $\|\mathbf{Q}'\|_2^2 - \frac{1}{M}\|\widehat{\mathbf{Q}'}\mathbf{1}_S\|_2^2 \geq 3\epsilon^2\left(\frac{m'}{m}\right)^2 + \frac{1}{m'}$ **then return** reject
  7: **else**
  8:     **return** $\widehat{\mathbf{H}'} = (\widehat{\mathbf{Q}'}(\xi))_{\xi\in S}$
  9: **end if**

---

The detailed proof of Theorem 6 is given in the full version.

### 3.1 A Tolerant $L_2$ Tester for Identity to a Pseudodistribution

As previously mentioned, one building block in the proof of Theorem 6 (and a result that may be of independent interest) is an optimal $L_2$ identity testing algorithm. Our tester and its analysis are very similar to the tolerant $L_2$ closeness testing algorithm of Chan et al. [14], with the obvious simplifications pertaining to identity (instead of closeness). The main difference is that we emphasize here the fact that $\mathbf{P}^*$ need not be an actual distribution: any $\mathbf{P}^*\colon[r]\to\mathbb{R}$ would do, even taking negative values. This will turn out to be crucial for our applications.

**Proposition 1.** *There exists an absolute constant $c>0$ such that the above algorithm (Algorithm 2), when given $\mathrm{Poi}(m)$ samples drawn from a distribution $\mathbf{P}$ and an explicit function $\mathbf{P}^*\colon[r]\to\mathbb{R}$ will,*

---

**Algorithm 2** Tolerant $L_2$ identity tester

---

**Require:** $\epsilon \in (0, 1)$, $\mathrm{Poi}(m)$ samples from distributions $\mathbf{P}$ over $[r]$, with $X_i$ denoting the number of occurrences of the $i$-th domain elements in the samples from $\mathbf{P}$, and $\mathbf{P}^*$ being a fixed, known pseudo distribution over $[r]$.

**Ensure:** Returns accept if $\|\mathbf{P} - \mathbf{P}^*\|_2 \leq \epsilon$ and reject if $\|\mathbf{P} - \mathbf{P}^*\|_2 \geq 2\epsilon$.

    Define $Z = \sum_{i=1}^{r}(X_i - m\mathbf{P}^*(i))^2 - X_i$.                  ▷ Can actually be computed in $O(m)$ time

    Return reject if $\frac{\sqrt{Z}}{m} > \sqrt{3}\epsilon$, accept otherwise.

---

*with probability at least $3/4$, distinguishes between* **(a)** $\|\mathbf{P} - \mathbf{P}^*\|_2 \leq \epsilon$ *and* **(b)** $\|\mathbf{P} - \mathbf{P}^*\|_2 \geq 2\epsilon$ *provided that $m \geq c\frac{\sqrt{b}}{\epsilon^2}$, where $b$ is an upper bound on $\|\mathbf{P}\|_2^2, \|\mathbf{P}^*\|_2^2$.*

*Moreover, we have the following stronger statement: in case (a), the statistic $Z$ computed in the algorithm satisfies $\frac{\sqrt{Z}}{m} \leq \sqrt{2.9}\epsilon$ with probability at least $3/4$, while in case (b) we have $\frac{\sqrt{Z}}{m} \geq \sqrt{3.1}\epsilon$ with probability at least $3/4$.*

## 4   The General tester

In this section, we provide our general testing framework. In more detail, our theorem (Theorem 7) has the following flavor: if $\mathcal{P}$ is a property of distributions such that every $\mathbf{P} \in \mathcal{P}$ has both (i) small effective support and (ii) sparse effective Fourier support, then one can test membership to $\mathcal{P}$ with $O(\sqrt{sM}/\epsilon^2 + s/\epsilon^2)$ samples (where $M$ and $s$ are the bounds on the effective support and effective Fourier support, respectively). As a caveat, we do require that the sparse effective Fourier support $S$ be independent of $\mathbf{P} \in \mathcal{P}$, i.e., is a characteristic of the class $\mathcal{P}$ itself.

The high-level idea is then quite simple: the algorithm proceeds in three stages, namely the *effective support test*, the *Fourier effective support test*, and the *projection step*. In the first, it takes some samples from $\mathbf{P}$ to identify what should be the effective support $I$ of $\mathbf{P}$, if $\mathbf{P}$ did have the property: and then checks that indeed $|I| \leq M$ (as it should) and that $\mathbf{P}$ puts probability mass $1 - O(\epsilon)$ on $I$. In the second stage, it invokes the Fourier testing algorithm of Section 3 to verify that $\widehat{\mathbf{P}}$ indeed puts very little Fourier mass outside of $S$; and, having verified this, learns very accurately the set of Fourier coefficients of $\mathbf{P}$ on this set $S$, in $L_2$ distance. At this point, either the algorithm has detected that $\mathbf{P}$ violates some required characteristic of the distributions in $\mathcal{P}$, in which case it has rejected already; or is guaranteed to have *learned* a good approximation $\mathbf{H}$ of $\mathbf{P}$, by the Fourier learning performed in the second stage. It only remains to perform the third stage, which "projects" this good approximation $\mathbf{H}$ of $\mathbf{P}$ onto $\mathcal{P}$ to verify that $\mathbf{H}$ is close to some distribution $\mathbf{P}^* \in \mathcal{P}$ (as it should if indeed $\mathbf{P} \in \mathcal{P}$).

**Theorem 7** (General Testing Statement). *Assume $\mathcal{P} \subseteq \Delta(\mathbb{N})$ is a property of distributions satisfying the following. There exist $S \colon (0, 1] \to 2^{\mathbb{N}}$, $M \colon (0, 1] \to \mathbb{N}$, and $q_I \colon (0, 1] \to \mathbb{N}$ such that, for every $\epsilon \in (0, 1]$,*

1. *Fourier sparsity: for all $\mathbf{P} \in \mathcal{P}$, the Fourier transform (modulo $M(\epsilon)$) of $\mathbf{P}$ is concentrated on $S(\epsilon)$: namely, $\left\|\widehat{\mathbf{P}}\mathbb{1}_{\overline{S(\epsilon)}}\right\|_2^2 \leq \frac{\epsilon^2}{100}$.*

2. *Support sparsity: for all $\mathbf{P} \in \mathcal{P}$, there exists an interval $I(\mathbf{P}) \subseteq \mathbb{N}$ with $|I(\mathbf{P})| \leq M(\epsilon)$ such that (i) $\mathbf{P}$ is concentrated on $I(\mathbf{P})$: namely, $\mathbf{P}(I(\mathbf{P})) \geq 1 - \frac{\epsilon}{5}$ and (ii) $I(\mathbf{P})$ can be identified with probability at least $19/20$ from $q_I(\epsilon)$ samples from $\mathbf{P}$.*

3. *Projection: there exists a procedure $\mathrm{PROJECT}_{\mathcal{P}}$ which, on input $\epsilon \in (0, 1]$ and the explicit description of a distribution $\mathbf{H} \in \Delta(\mathbb{N})$, runs in time $T(\epsilon)$; and outputs accept if $d_{\mathrm{TV}}(\mathbf{H}, \mathcal{P}) \leq \frac{2\epsilon}{5}$, and reject if $d_{\mathrm{TV}}(\mathbf{H}, \mathcal{P}) > \frac{\epsilon}{2}$ (and can answer either otherwise).*

4. *(Optional) $L_2$-norm bound: there exists $b \in (0, 1]$ such that, for all $\mathbf{P} \in \mathcal{P}$, $\|\mathbf{P}\|_2^2 \leq b$.*

*Then, there exists a testing algorithm for $\mathcal{P}$, in the usual standard sense: it outputs either accept or reject, and satisfies the following.*

1. *if $\mathbf{P} \in \mathcal{P}$, then it outputs accept with probability at least $3/5$;*

---

**Algorithm 3** Algorithm `Test-Fourier-Sparse-Class`

---

**Require:** sample access to a distribution $\mathbf{P} \in \Delta(\mathbb{N})$, parameter $\epsilon \in (0,1]$, $b \in (0,1]$, functions $S\colon (0,1] \to 2^{\mathbb{N}}$, $M\colon (0,1] \to \mathbb{N}$, $q_I\colon (0,1] \to \mathbb{N}$, and procedure PROJECT$_{\mathcal{P}}$ as in Theorem 7

1: Effective Support
2:     Take $q_I(\epsilon)$ samples from $\mathbf{P}$ to identify a "candidate set" $I$.      ▷ Guaranteed to work w.p. 19/20 if $\mathbf{P} \in \mathcal{P}$.
3:     Draw $O(1/\epsilon)$ samples from $\mathbf{P}$, to distinguish between $\mathbf{P}(I) \geq 1 - \frac{\epsilon}{5}$ and $\mathbf{P}(I) < 1 - \frac{\epsilon}{4}$. ▷ Correct w.p. 19/20.
4:     **if** $|I| > M(\epsilon)$ or we detected that $\mathbf{P}(I) > \frac{\epsilon}{4}$ **then**
5:         **return** reject
6:     **end if**
7:
8: Fourier Effective Support
9:     Simulating sample access to $\mathbf{P}' \overset{\text{def}}{=} \mathbf{P} \bmod M(\epsilon)$, call Algorithm 1 on $\mathbf{P}'$ with parameters $M(\epsilon)$, $\frac{\epsilon}{5\sqrt{M(\epsilon)}}$, $b$, and $S(\epsilon)$.
10:     **if** Algorithm 1 returned reject **then**
11:         **return** reject
12:     **end if**
13:     Let $\widehat{\mathbf{H}} = (\widehat{\mathbf{H}}(\xi))_{\xi \in S(\epsilon)}$ denote the collection of Fourier coefficients it outputs, and $\mathbf{H}$ their inverse Fourier transform (modulo $M(\epsilon)$)      ▷ Do not actually compute $\mathbf{H}$ here.
14:
15: Projection Step
16:     Call PROJECT$_{\mathcal{P}}$ on parameters $\epsilon$ and $\mathbf{H}$, and **return** accept if it does, reject otherwise.
17:

---

    2. *if $d_{\mathrm{TV}}(\mathbf{P}, \mathcal{P}) > \epsilon$, then it outputs* reject *with probability at least $3/5$.*

*The algorithm takes*

$$
O\left( \frac{\sqrt{|S(\epsilon)|M(\epsilon)}}{\epsilon^2} + \frac{|S(\epsilon)|}{\epsilon^2} + q_I(\epsilon) \right)
$$

*samples from $\mathbf{P}$ (if Item 4 holds, one can replace the above bound by $O\left( \frac{\sqrt{b}M(\epsilon)}{\epsilon^2} + \frac{|S(\epsilon)|}{\epsilon^2} + q_I(\epsilon) \right)$); and runs in time $O(m|S| + T(\epsilon))$, where $m$ is the sample complexity.*

*Moreover, whenever the algorithm outputs* accept*, it also learns $\mathbf{P}$; that is, it provides a hypothesis $\mathbf{H}$ such that $d_{\mathrm{TV}}(\mathbf{P}, \mathbf{H}) \leq \epsilon$ with probability at least $3/5$.*

We remark that the statement of Theorem 7 can be made slightly more general; specifically, one can allow the procedure PROJECT$_{\mathcal{P}}$ to have sample access to $\mathbf{P}$ and err with small probability, and further provide it with the Fourier coefficients $\widehat{\mathbf{H}}$ learnt in the previous step.

## 5   Lower Bound for PMD Testing

In this section, we obtain a lower bound to complement our upper bound for testing Poisson Multinomial Distributions. Namely, we prove the following:

**Theorem 8.** *There exists absolute constants $c, c' \in (0,1)$ such that the following holds. For any $k \leq n^c$ and $\epsilon \geq 1/2^{c'n}$, any testing algorithm for the class of $\mathcal{PMD}_{n,k}$ must have sample complexity $\Omega\left( \left( \frac{4\pi}{k} \right)^{k/4} \frac{n^{(k-1)/4}}{\epsilon^2} \right)$.*

The proof will rely on the lower bound framework of [11], reducing testing $\mathcal{PMD}_{n,k}$ to testing identity to some suitable hard distribution $\mathbf{P}^* \in \mathcal{PMD}_{n,k}$. To do so, we need to (a) choose a convenient $\mathbf{P}^* \in \mathcal{PMD}_{n,k}$; (b) prove that testing identity to $\mathbf{P}^*$ requires that many samples (we shall do so by invoking the [54] instance-by-instance lower bound method); (c) provide an agnostic learning algorithm for $\mathcal{PMD}_{n,k}$ with small enough sample complexity, for the reduction to go through. Invoking [11, Theorem 18] with these ingredients will then conclude the argument.

## Footnotes

[1]The full version of this paper is available at [13].

[2]By the term "non-trivial" here we refer to a testing algorithm that uses fewer samples than just learning the unknown distribution and then checking whether it is close to a distribution in the family.

[3]Here, we use the notation $\Omega_k(\cdot)$, $O_k(\cdot)$ to indicate that the parameter $k$ is seen as a constant.

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
