[Reviews · NeurIPS 2018]

Reviewer 1



This paper presents a general method for distribution testing for classes of distributions that have a sparse discrete Fourier transform representation. As specific applications of the general method, the sums of independent integer random variables, the Poisson multinomial distribution, and the discrete log-concave distribution are considered, and it is shown that for all three applications, the proposed method can improve previously existing upper bounds for sample complexity. The proposed method is based on a test that takes as input a subset S and sample access to an unknown distribution Q and tests if the distribution Q has DFT (approximately) concentrated on S or not. The proposed algorithm to do this is simple, it computes the DFT of the empirical distribution and checks two things: 1. Does the empirical distribution have a sufficiently small energy outside S? 2. Does the DFT of empirical distribution approximate well that of the original distribution on S? The sample complexity for both these things to happen is analyzed by showing that the energy in the approximation error between Q and a function H obtained by restricting the empirical distribution to S in the frequency domain equals to the energies in the two error terms in items above. Therefore, when this approximation error is small, each error in the two items above is simultaneously small. Finally, to handle the sample complexity required to make this error small the paper uses a simple result on L_2 testing (Actually they build on a much stronger result, but actually need a much simpler implication of it.) The paper has presented this as a meta tool that can be applied to the three problems specified above. But applying this tool to these problems is not straightforward. For each, tedious analysis is required to make the meta procedure above useful and compatible with known tests for these cases. Overall, I think that the analysis of the general procedure as a meta tool, which is projected as the major component of the paper in the abridged 10 page version, is not so difficult. It is simply based on an earlier work [32]. What is really complicated is applying this tool to each problem, a procedure that has been described in the supplementary material. While at a high level, the analysis in the paper is based on simply combining the known methods for these problems with the meta procedure. However, making this combination work requires an insight into the structure of each of these problems. I feel this is the strength of this paper. It has successfully applied a simple approach to many problem, but the justification is technically tedious. Thus, a weakness of this paper is that the ideas pursued follow easily from prior work (discounting the technical difficulties required to complete a formal proof). Overall, I feel that it is strong paper in theory of distribution testing, but may not add much to the algorithms themselves, which are obvious from prior work.

Reviewer 2



The paper proposes a generic method for testing whether a distribution belongs to a family of distributions, or it is far from it (w.r.t. TV distance) This generic method works when the Fourier spectrums of distributions in the family are sparse. The authors then show the corollaries for learning certain families of distributions, namely SIIRVs, PBDs, PMDs, and Log-concave distributions. The authors argue that the provided results improve over the (trivial) learning by testing approach. While this seems to be true for SIIRV (improving from n^(1/2) to n^(1/4) in sample complexity), it does not seem to be true for these two other cases: + Testing Log-concavity (Theorem 4) does not improve over the (trivial) testing by learning paradigm, using the known efficient learners for log-concave distributions [4,5,6] + Testing PMDs (Theorem 3) does not improve much over the (trivial) testing by learning paradigm, using the efficient learning algorithm of [2]. (both will give (nk)^(O(k)) Please clarify if I am wrong. Otherwise, I think the current write-up is misleading (I am surprised the authors do not mention this). The overall idea of exploiting the sparsity of Fourier spectrum is reminiscent of [2] and [3]. Although these papers do not address testing specifically, they are similar in techniques used. [1] I. Diakonikolas, D. M. Kane, and A. Stewart. Properly learning Poisson binomial distributions in almost polynomial time, COLT 2016 [2] I. Diakonikolas, D. M. Kane, and A. Stewart. The Fourier Transform of Poisson Multinomial Distributions and its Algorithmic Applications. STOC 2016 [3] I. Diakonikolas, D. M. Kane, and A. Stewart. Optimal Learning via the Fourier Transform for Sums of Independent Integer Random Variables. COLT 2016 [4] Chan, S. O., Diakonikolas, I., Servedio, R. A., & Sun, X. Efficient density estimation via piecewise polynomial approximation. STOC 2014 [5] Acharya, J., Diakonikolas, I., Li, J., & Schmidt, L. Sample-optimal density estimation in nearly-linear time. SODA 2017 This one seems to be a 2016 Arxiv preprint, but I use it because the authors themselves have cited to it. [6] I. Diakonikolas, D. M. Kane, and A. Stewart. Efficient robust proper learning of log-concave distributions. CoRR, abs/1606.03077, 2016. ==== After reading the authors response, I changed my decision to accept. As the author mentioned, the testing by learning paradigm requires a tolerant testing (rather than identity testing) after the learning step. So the trivial upper bounds that I was mentioning don't work.

Reviewer 3



Summary The paper studies the problem of testing membership to a family of distributions. In this problem the goal is to decide if an unknown distribution belongs in some pre-specified class of distributions or if it is at least eps-far in total variation distance from any distribution in the class. While the problem of testing identity to a single distribution is well understood, testing membership to a class is significantly more challenging. This paper proposes a novel framework for testing membership to classes whose Fourier spectrum is sufficiently concentrated over some small domain. The main idea of the work is that in such cases it is possible to learn the distribution with few samples not only in total variation distance eps but also in L2 distance eps/sqrt(N) where N is the size of the effective support. This allows applying a testing via learning approach that does not work for TV distance as it requires a tolerant identity tester. In contrast, tolerant testing for L2 distance is known to be significantly more sample efficient. The paper applies the framework to families of distributions that can be expressed as sums of independent random variables and log-concave distributions. The sampling complexities that are achieved are: - Poisson Binomial Distributions (PBD): ~n^(1/4)/eps^2 - Sums of k Integer Random Variables (kSIIRV): ~(kn^(1/4) + k^2)/eps^2 - Poisson Multinomial Distributions (PMD): ~n^((k-1)/4)/eps^2 - Log-Concave Distributions: ~sqrt(n)/eps^2 + 1/eps^2.5 The first result and fourth results are improvements over prior work removing additive terms of the sample complexity in the order of 1/eps^6 and 1/eps^5 respectively. The achieved results are nearly tight. Prior work has established that ~n^(1/4)/eps^2, sqrt(k) n^(1/4)/eps^2 are necessary for the cases of PBD and kSIIRV respectively, while a sqrt(n)/eps^2 lower bound follows from lower bounds for uniformity testing. In addition, this work provides a ~n^((k-1)/4)/eps^2 lower bound for the case of PMD. Comments The proposed method based on Fourier Transform is quite interesting and the paper nicely illustrate its power and applicability through a range of different settings obtaining near optimal sample complexity bounds that qualitatively match the information theoretic lower bounds. An important weakness of this work is its presentation. There is very little content in the main body of the paper. Apart from the framework, no detail is given about how it is applied in the results. The explanation in the introduction is quite vague and only mentions difficulties in applying the framework but not how it is applied. Right now everything is pushed to the supplementary materials which is highly technical and hard to read. It should be possible to give a more concrete outline with simplified versions of the algorithms in the main body of how the framework is applied and what Fourier properties of the families of distributions are used. I understand that the NIPS format gives limited space but I find that its use was not well optimized. Another point missing in the paper is a unified summary of related work. There has been a lot of directly related work for testing families of distributions which is only partially invoked when comparing the results of the paper. Recommendation Overall, I believe that the proposed framework is quite interesting and novel, and is able to improve on prior work of the literature and get near optimal bounds to many interesting problems. The main drawback I find is the presentation of the results. The current submission contains very little content in the main body. I think an outline of the algorithms and how the framework is applied to the families of distributions considered are necessary to be included.